# Taxonomy and Phylogeny of the *Favolaschia calocera* Complex (Mycenaceae) with Descriptions of Four New Species

Qiu-Yue Zhang  and Yu-Cheng Dai *

Institute of Microbiology, School of Ecology and Nature Conservation, Beijing Forestry University, Beijing 100083, China; zhangqiuyue@bjfu.edu.cn
* Correspondence: yuchengd@yahoo.com

**Abstract:** *Favolaschia calocera* was originally described from Madagascar, and reported to have a worldwide distribution. In the current study, samples of the *Favolaschia calocera* from Central America, Australia, China, Kenya, Italy, New Zealand, and Thailand were analyzed by using both morphological and molecular methods. Phylogenetic analyses were based on the internal transcribed spacer (ITS) dataset, and the combined five-locus dataset of ITS, large subunit nuclear ribosomal RNA gene (nLSU), the small subunit mitochondrial rRNA gene (mt-SSU), the small subunit of nuclear ribosomal RNA gene (nu-SSU), and the translation elongation factor 1α (TEF1). Our study proves that *Favolaschia calocera* is a species complex, and six species are recognized in the complex including four new species. Three new species *F. brevibasidiata*, *F. brevistipitata*, and *F. longistipitata* from China; and one new species *F. minutissima* from Asia. In addition, *Favolaschia claudopus* (Singer) Q.Y. Zhang & C. Dai, earlier treated as a variety of *Favolaschia calocera* R. Heim, were raised to species rank. Illustrated descriptions of these five new taxa are given. An identification key and a comparison of the characteristics of species in the *Favolaschia calocera* complex are provided.

**Keywords:** multi-gene phylogeny; Mycenaceae; species complex; taxonomy

## 1. Introduction

*Favolaschia* (Pat.) Pat. (Mycenaceae, Agaricales) was established based on *F. gaillardii* (Pat.) Pat. [1]. It is characterized by poroid hymenophore, a monomitic hyphal system, gelatinous hyphal structure, the presence of gloeocystidia and acanthocystida, and amyloid basidiospores [1–5]. *Favolaschia* produces basidiomes on dead plant material, but they may have a biotrophic phase as found in *Mycena* [6,7], since many are known to be host specific. More than 100 taxa have been recorded in *Favolaschia*, although it is suspected that these taxa represent only around 50 species [8–10]. The genus has a worldwide distribution (excluding Antarctica), with a high diversity in South America [5].

*Favolaschia calocera* R. Heim was first described from Madagascar [11]. It differs from other species in the genus by its bright orange or yellow basidiocarps with a distinct laterally stipe and numerous gloeocystidia and acanthocysts. In addition, *Favolaschia calocera* is a conspicuous fungus, and it is known to occur in different environments, such as forests or shrub-lands, non-forested rural areas, home or public gardens etc., and occurs on over 50 different plant species, such as decaying dicotyledonous tree trunks, logs, and branches [12,13].

*Favolaschia calocera* has always been considered to be an invasive fungus by several countries [14,15]. Outside its type locality in Madagascar, it was found in New Zealand and Italy in 1969 and 1999 [13,16]. Johntson et al. [12,13] mentioned that *Favolaschia calocera* spread to Australia and New Zealand presumably by shipped timber. Vizzini and Zotti [16] believed that *Favolaschia calocera* in Norfolk Island was introduced from New Zealand. Over the past couple of decades, it has spread into Europe. In the process of the migration, the morphology and molecules of *Favolaschia calocera* gradually diverged. As early as in 1974, Singer [8] reported the morphological differences of *Favolaschia calocera* collections

from Madagascar and New Zealand. Based on molecular phylogeographic analysis, Vizzini et al. [14] and Ainsworth et al. [17] indicated *Favolaschia calocera* is divided into two main groups: the European and Oceania samples clustered in one group, and the American and Asian materials formed another group. Until recently, there has been no systematic study on the distribution of this species in China. Our initial investigation found that *Favolaschia calocera* is common in tropical China, and displays a higher genetic variability.

The aim of this study was to investigate the diversity and phylogeny within the *Favolaschia calocera* species complex, and five new taxa were described. The key characteristics for distinguishing species are provided.

## 2. Materials and Methods

### 2.1. Morphological Studies

Studied specimens collected by authors or new collections from China and Australia are deposited at the herbarium of the Institute of Microbiology, Beijing Forestry University (BJFC). Sections of basidiocarps were studied microscopically according to Cui et al. [18] using a Nikon Eclipse 80i microscope with phase contrast illumination. Color terms are cited from Petersen [19]. In presenting spore size variation, 5% of measurements were excluded from each end of the range and this value is given in parentheses. The following abbreviations are used in the description: KOH = 5% potassium hydroxide; IKI = Melzer's reagent; IKI+ = amyloid; CB = Cotton Blue; CB− = acyanophilous; L = arithmetic average of all spores; W = arithmetic average of all spores; Q = the L/W ratio; *n* = number of spores measured from the given number of specimens.

### 2.2. DNA Extraction and Sequencing

A CTAB rapid plant genome extraction kit (Aidlab Biotechnologies, Co., Ltd., Beijing, China) was used to obtain PCR products from dried specimens, following the manufacturer's instructions with some modifications [20]. To generate PCR amplicons, the following primer pairs were used: ITS5 and ITS4 for ITS, and LR0R and LR7 for nLSU, MS1 and MS2 for mt-SSU, NS1 and NS4 for nu-SSU, and 983F and 1567R for TEF1 [21,22].

The PCR cycling schedules for different DNA sequences used in this study followed those used in Zhou et al. and Liu et al. [23,24] with some modifications. The PCR procedure for ITS, mt-SSU, and TEF1 was as follows: initial denaturation at 95 °C for 3 min, followed by 35 cycles at 94 °C for 40 s, 54 °C for 45 s, and 72 °C for 1 min, and a final extension of 72 °C for 10 min. The PCR procedure for nLSU was as follows: initial denaturation at 94 °C for 1 min, followed by 35 cycles at 94 °C for 30 s, 50 °C for 1 min and 72 °C for 1.5 min, and a final extension of 72 °C for 10 min. The PCR procedure for the nu-SSU was as follows: initial denaturation at 94 °C for 1 min, followed by 34 cycles at 94 °C for 30 s, 53 °C for 1 min, and 72 °C for 1.5 min, and a final extension of 72 °C for 10 min. The PCR products were purified and sequenced at the Beijing Genomics Institute (BGI), China with the same primers and the sequences were deposited in GenBank and are listed in Table 1.

### 2.3. Phylogenetic Analyses

Besides the newly generated sequences for this study, other related sequences from GenBank were also included in the phylogenetic analysis. *Panellus stipticus* (Bull.) P. Karst. is used as the outgroup in Figure 1 [26], and *Favolaschia varariotecta* Singer is used as the outgroup in Figure 2 [14]. The sequences used ClustalX [27] and were manually adjusted in BioEdit [28]. Trees were shown in TreeView 1.6.6 [29].

**Table 1.** A list of species, specimens, and GenBank accession numbers of the sequences used in this study.

| Species | Sample No. | Location | GenBank Accession No. | | | | | Reference |
|---------|-----------|----------|------|------|--------|--------|------|-----------|
| | | | **ITS** | **nLSU** | **mt-SSU** | **Nu-SSU** | **TEF1** | |
| *Favolaschia andina* | KG0025 | Panama | HM246678 | — | — | — | — | [5] |
| *F. austrocyatheae* | PDD75609 | New Zealand | NR132809 | — | — | — | — | [13] |
| *F. austrocyatheae* | PDD75609 | New Zealand | DQ026257 | — | — | — | — | [13] |
| *F. aurantiaca* | FK2047 | Brazil | JX987670 | — | — | — | — | [25] |
| *F. auriscalpium* | | | KY649461 | — | — | — | — | — |
| **F. brevibasidiata** | **Cui 6573** | **Hainan, China** | **MZ661794** | | **MZ661749** | **MZ661765** | **MZ959379** | **Present study** |
| *F. brevibasidiata* | JM98186 | Yunnan, China | DQ026239 | — | — | —— | — | [13] |
| **F. brevistipitata** | **Dai 19780** | **Yunnan, China** | **MZ661772** | **MZ661742** | **MZ661753** | **MZ661769** | **MZ959383** | **Present study** |
| **F. brevistipitata** | **Dai 19855** | **Yunnan, China** | **MZ661773** | **MZ661743** | **MZ661754** | **MZ661770** | **MZ959384** | **Present study** |
| **F. brevistipitata** | **Dai 19856** | **Yunnan, China** | **MZ661774** | **MZ661744** | **MZ661755** | **MZ661771** | **MZ959385** | **Present study** |
| *F. calocera* | PC99060 | Madagascar | DQ26252 | — | — | — | — | [13] |
| *F. calocera* | PC99497 | Madagascar | DQ026253 | — | — | — | — | [13] |
| *F. cinnabarina* | 4421 | Brazil | JX987669 | — | — | — | — | [25] |
| *F. cinnabarina* | DUKE4039 | Puerto Rico | DQ026242 | — | — | — | — | [13] |
| *F. cyatheae* | PDD75316 | New Zealand | NR132808 | — | — | — | — | [13] |
| *F. cyatheae* | PDD75316 | New Zealand | DQ026256 | — | — | — | — | [13] |
| **F. claudopus** | **Dai 18656** | **Australia** | **MZ661775** | **MZ661735** | **MZ661745** | **MZ661761** | **—** | **Present study** |
| **F. claudopus** | **Dai 18663** | **Australia** | **MZ661776** | **MZ661734** | | | **MZ959375** | **Present study** |
| *F. claudopus* | SR346 | Kenya | DQ026237 | — | — | — | — | [13] |
| *F. claudopus* | PDD74554 | New Zealand | DQ026251 | — | — | — | — | [13] |
| *F. claudopus* | PDD75323 | New Zealand | DQ026248 | — | — | — | — | [13] |
| *F. claudopus* | PDD75686 | New Zealand | DQ026249 | — | — | — | — | [13] |
| *F. claudopus* | DUKE2952 | New Zealand | DQ026238 | — | — | — | — | [13] |
| *F. claudopus* | GDOR 03102001 | Italy | EU489633 | — | — | — | — | [14] |
| *F. claudopus* | GDOR 05100901 | Italy | EU489638 | — | — | — | — | [14] |
| **F. longistipitata** | **Dai 13221** | **Yunnan, China** | **MZ661777** | **—** | **—** | **—** | **—** | **Present study** |
| **F. longistipitata** | **Dai 13226** | **Yunnan, China** | **MZ661778** | **—** | **—** | **—** | **—** | **Present study** |
| **F. longistipitata** | **Cui 11128** | **Yunnan, China** | **MZ661779** | **—** | **—** | **—** | **—** | **Present study** |
| **F. longistipitata** | **Dai 17597** | **Yunnan, China** | **MZ661780** | **—** | **—** | **—** | **—** | **Present study** |
| **F. longistipitata** | **Dai 17598** | **Yunnan, China** | **MZ661781** | **—** | **—** | **—** | **—** | **Present study** |
| **F. longistipitata** | **Dai 17601** | **Yunnan, China** | **MZ661782** | **—** | **—** | **—** | **—** | **Present study** |
| **F. longistipitata** | **Dai 19781** | **Yunnan, China** | **MZ661783** | **—** | **—** | **—** | **—** | **Present study** |
| **F. longistipitata** | **Dai 19799** | **Yunnan, China** | **MZ661784** | **MZ661739** | **MZ661750** | **MZ661766** | **MZ959380** | **Present study** |
| **F. longistipitata** | **Dai 19893** | **Yunnan, China** | **MZ661785** | **MZ661740** | **MZ661751** | **MZ661767** | **MZ959381** | **Present study** |
| **F. longistipitata** | **Dai 20019** | **Yunnan, China** | **MZ661786** | **MZ661741** | **MZ661752** | **MZ661768** | **MZ959382** | **Present study** |
| **F. longistipitata** | **Dai 20328** | **Yunnan, China** | **MZ661787** | **—** | **—** | **—** | **—** | **Present study** |
| **F. longistipitata** | **Dai 20341** | **Yunnan, China** | **MZ661788** | **—** | **—** | **—** | **—** | **Present study** |
| **F. longistipitata** | **Dai 20355** | **Yunnan, China** | **MZ661789** | **—** | **—** | **—** | **—** | **Present study** |
| *F. luteoaurantiaca* | 4475 | — | JX987667 | — | — | — | — | [25] |
| *F. luteoaurantiaca* | SP445750 | Brazil | NR132874 | — | — | — | — | [25] |
| *F. macropora* | KG0027 | Panama | NR132845 | — | — | — | — | [5] |
| *F. minutissima* | JM98372 | Thailand | DQ026240 | — | — | — | — | [13] |
| *F. minutissima* | Dai 10753 | Hainan, China | MZ661790 | | | | | |
| **F. minutissima** | **Dai 20085** | **Hainan, China** | **MZ661791** | **MZ661736** | **MZ661746** | **MZ661762** | **MZ959376** | **Present study** |
| **F. minutissima** | **Dai 20086** | **Hainan, China** | **MZ661792** | **MZ661737** | **MZ661748** | **MZ661764** | **MZ959378** | **Present study** |
| **F. minutissima** | **Dai 20088** | **Hainan, China** | **MZ661793** | **MZ661738** | **MZ661747** | **MZ661763** | **MZ959377** | **Present study** |
| *F. peziziformis* | ICMP1575 | Japan | DQ026255 | — | — | — | — | [13] |
| *F. pustulosa* | DUKE3316 | Papua New Guinea | DQ026245 | — | — | — | — | [13] |
| *F. pustulosa* | PDD75686 | New Zealand | DQ026254 | — | — | — | — | [13] |
| *F. sp* | 4550 | Panama | JX987668 | — | — | — | — | [5] |
| *F. sp* | DUKE2708 | Australia | DQ026234 | — | — | — | — | [13] |
| *F. sp* | DUKE2876 | Australia | DQ026235 | — | — | — | — | [13] |
| *F. sp* | DUKE3195 | Papua New Guinea | DQ026236 | — | — | — | — | [13] |
| *F. sprucei* | TH 6418 | Guyana | DQ026246 | — | — | — | — | [13] |
| *F. tonkinensis* | JM98229 | Yunnan, China | DQ026247 | — | — | — | — | [13] |
| *F. varariotecta* | DUKE3893 | Puerto Rico | DQ026243 | — | — | — | — | [13] |
| *F. varariotecta* | DUKE4038 | Puerto Rico | DQ026244 | — | — | — | — | [13] |
| *Panellus stypticus* | TN 4319 | Switzerland | AF289069 | — | — | — | — | [26] |

New sequences are shown in bold.

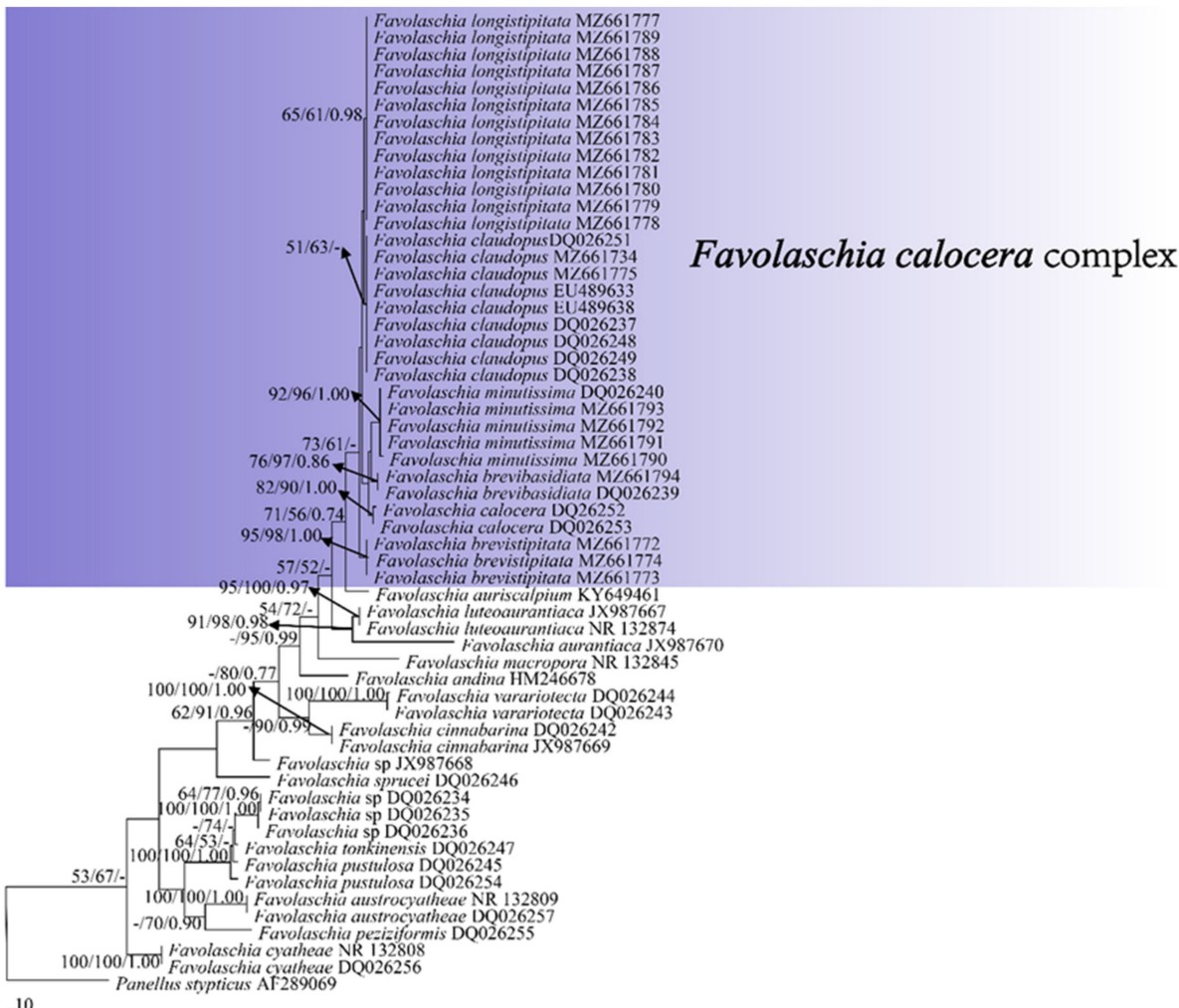

**Figure 1.** Maximum parsimony tree illustrating the phylogeny of the *Favolaschia calocera* complex and related group based on the combined sequences dataset of ITS. Branches are labelled with parsimony bootstrap values higher than 50%, and Bayesian posterior probabilities more than 0.70.

Maximum parsimony (MP) analysis was used for the ITS and the ITS + nLSU + mt-SSU + nu-SSU + TEF1 datasets in PAUP* 4.0b10 [30]. Trees were generated using 100 replicates of random stepwise addition of sequence and tree-bisection reconnection (TBR) branch-swapping algorithm, with all characters given equal weight. Branch supports for all parsimony analyses were estimated by performing 1000 bootstrap (BT) replicates [31]. Descriptive tree statistics, tree length (TL), consistency index (CI), retention index (RI), rescaled consistency index (RC), and homoplasy index (HI) were calculated for each maximum parsimonious tree generated.

RAxML 7.2.8 (GitHub, San Francisco, CA, USA) was used for maximum likelihood (ML) analysis. The default settings of the GTR+I+G model were used for all parameters in the ML analysis [32]. The branch support values were obtained using nonparametric bootstrapping with 1000 replicates [33].

Bayesian interference (BI) analyses were calculated with MrBayes 3.1.2 [34]. Four Markov chains were run for 5,000,000 generations until the split deviation frequency value was less than 0.01 and trees were sampled every 100 generations. Branches that received bootstrap support for BS (bootstrap support for MP and ML) values and BPPs (Bayesian posterior probabilities for BI) simultaneously not less than 50% and 0.70, respectively, were considered as significantly supported.

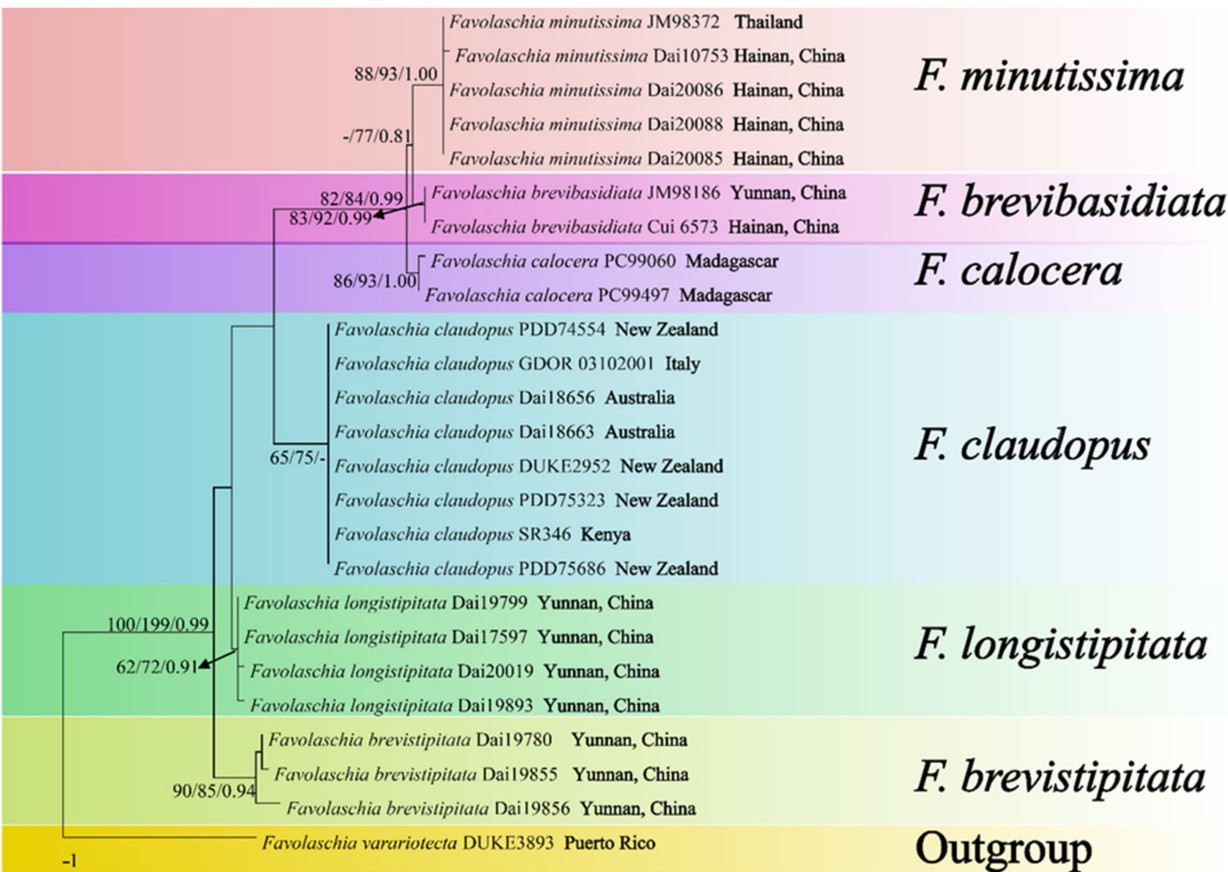

**Figure 2.** Maximum parsimony tree illustrating the phylogeny of the *Favolaschia calocera* complex based on ITS + nLSU + mt-SSU + nu-SSU + TEF1 sequences. Branches are labelled with parsimony bootstrap values higher than 50%, and Bayesian posterior probabilities more than 0.70.

## 3. Results

### 3.1. Molecular Phylogeny

The ITS base contained 57 sequences representing 23 taxa. The ITS dataset had an aligned length of 816 characters, of which 522 characters are constant, 122 characters are variable but parsimony uninformative, and 172 characters are parsimony informative. MP analysis yielded two equally parsimonious trees (TL = 542, CI = 0.675, RI = 0.847, RC = 0.572, HI = 0.325). BI and ML analyses produced consensus trees similar to MP analysis, and only the MP tree is shown. BI showed an average standard deviation of split frequencies = 0.006573.

The combined five-gene (ITS + nLSU + mt-SSU + nu-SSU + TEF1) dataset of 25 samples represented 7 taxa. It had an aligned length of 3470 characters, of which 3358 characters are constant, 65 characters are variable but parsimony uninformative, and 87 characters are parsimony informative. MP analysis yielded one tree (TL = 126, CI = 0.913, RI = 0.941, RC = 0.859, HI = 0.087). BI and ML analyses produced consensus trees similar to MP analysis, and only the MP tree is shown. BI showed an average standard deviation of split frequencies = 0.009683.

The phylogeny (Figure 1), based on ITS, shows that the five new taxa and *Favolaschia calocera* from Madagascar form six distinct lineages with robust support and cluster in the *F. calocera* complex clade. The phylogeny (Figure 2) based on ITS + nLSU + mt-SSU + nu-SSU + TEF1 results in a similar topology to the phylogeny based on ITS sequences, and is nested within the *Favolaschia calocera* complex, forming six distinct lineages.

*3.2. Taxonomy*

*Favolaschia brevibasidiata* Q.Y. Zhang & Y.C. Dai, sp. nov., Figures 3 and 4.
MycoBank: MB 841445.

**Diagnosis.** Differs from *F. brevistipitata*, *F. claudopus*, and *F. longistipitata* in color. Differs from *F. calocera* and *F. minutissima* in smaller basidia.

**Type.** CHINA, Hainan Province, Ledong County, Jianfengling Nature Reserve, 11 May 2009, Cui 6573 (holotype, BJFC004426).

**Etymology.** *Brevibasidiata* (Lat.): referring to the species having short basidia.

**Basidiocarps** annual, gregarious, gelatinous. **Pileus** 2–7 × 1.5–4 mm, orbicular, apricot-orange when fresh, cream buff when dry; pileal surface slightly undulated in a reticulate pattern matching the pores below, sometimes faintly pruinose when dry. **Hymenophore** concolorous with pileal surface, poroid, 50–120 pores per basidiocarp; pores 0.4–1 mm in diam, pentagonal to somewhat irregular or polygonal in shape, larger near the base and smaller near the edge, the marginal pores often incomplete, pore edges pruinose when dry. **Stipe** obvious, laterally attached, concolorous with pileus, cylindrical or tapered to a slightly wider base, sometimes curved, very finely velutinate under a lens, 2–8 mm long.

**Basidiospores** 10–12(–13.5) × 6–7.8(–8) μm, L = 11.33 μm, W = 6.9 μm, Q = 1.64 (*n* = 30/1), ellipsoid to broadly ellipsoid, hyaline, thin-walled, smooth, with some guttules, faintly IKI+, CB−. **Basidia** 23–30 × 8–10 μm, cylindric or clavate, few fusoid, contain some guttules, 2–spored, sterigmata 1.5–4 μm long; basidioles in shape similar to basidia, but slightly smaller. **Gloeocystidia** present at the edges of pores, in hymenium and pileipellis, slightly thick-walled, smooth, contents dense and yellow-orange, those in hymenium clavate to ventricose, 25–36 × 10–18 μm; those at the pore edges more or less the same size as those in pileipellis, clavate, cylindrical or subglobose, 13–36 × 10–22 μm. **Acanthocysts** at the edges of pores and in pileipellis, slightly thick-walled, contents dense and yellow-orange, clavate, short vesiculose to pyriform-elongated, 15–47 × 9–12 μm. **Tramal** hyphae subparallel along tubes, strongly gelatinized, partly branched, hyaline, thin-walled, some slightly inflated, 2–4(–6) μm in diameter. Pileipellis comprising a palisade of acanthocysts and gloeocystidia. Hyphae in stipe parallel along stipe, slightly thick-walled, some inflated, 3–6(–11) μm in diameter. Clamp connections absent.

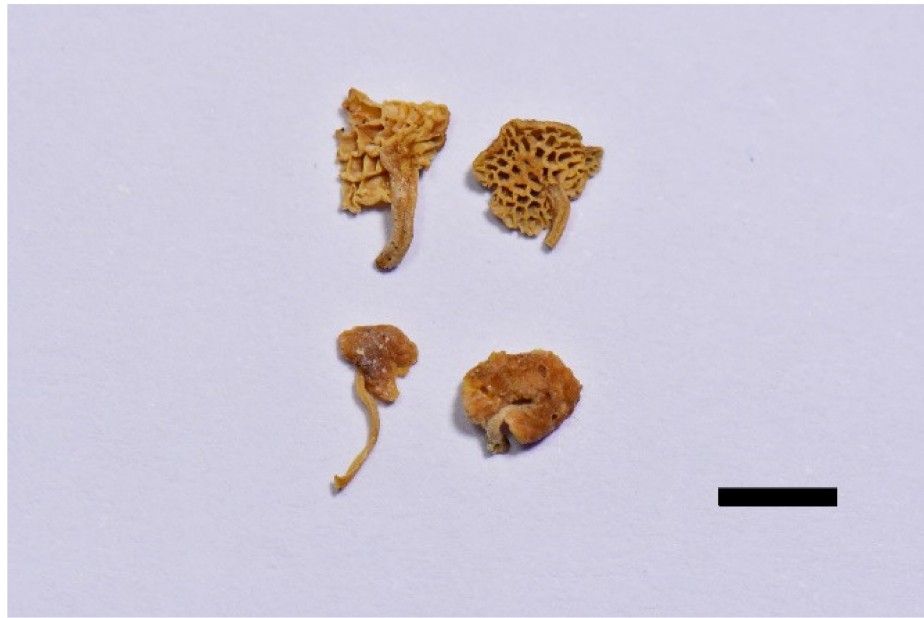

**Figure 3.** Dried basidiocarps of *Favolaschia brevibasidiata* (Holotype). Scale bar = 5 mm.

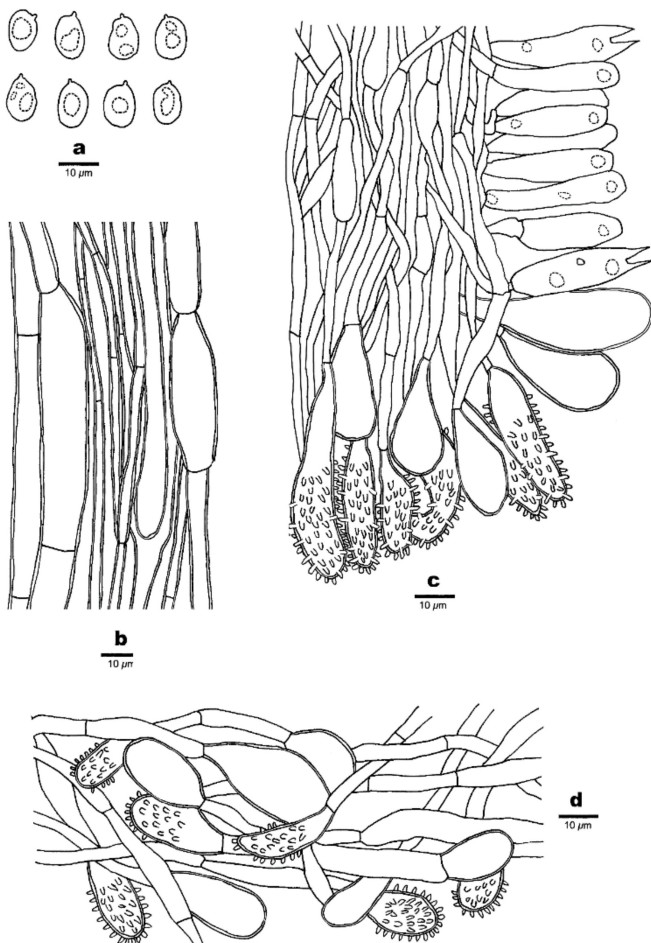

**Figure 4.** Microscopic structures of *Favolaschia brevibasidiata*. (Cui 6573, holotype). (**a**) Basidiospores. (**b**) Medullary hyphae of the stipe. (**c**) Hyphae at edge of pores showing gloeocystidia, acanthocystida, basidia and basidioles. (**d**) Hyphae of pileus cortex, composed of a hymeniform layer of acanthocysts and gloeocystidia.

*Favolaschia brevistipitata* Q.Y. Zhang & Y.C. Dai, sp. nov., Figures 5–7.
MycoBank: MB 841446.

**Diagnosis.** Differs from *F. brevibasidiata*, *F. calocera* and *F. minutissima* in color. Differs from *F. claudopus* and *F. longistipitata* in shorter stipes.

**Type.** CHINA, Yunnan Province, Pingbian County, Daweishan National Forest Park, fallen angiosperm branch, 26 June 2019, Dai 19780 (holotype, BJFC031455).

**Etymology.** *Brevistipitatus* (Lat.): referring to the species having a short stipe.

**Basidiocarps** annual, gregarious, gelatinous. **Pileus** 2–10 × 1–6 mm, reniform to subcircular, lemon-chrome when fresh, becoming curry yellow when dry; pileal surface slightly undulate in a reticulate pattern matching the pores below, sometimes faintly pruinose when dry. **Hymenophore** concolorous with pileal surface poroid, 50–180 pores per basidiocarp; pores 0.4–1.5 mm in diameter, pentagonal to hexagonal and sometimes irregularly elongated, larger near the base and smaller near the edge, the marginal pores often incomplete, pore edges pruinose when dry. **Stipe** often scarcely developed, but mostly present, laterally attached, concolorous with pileus, short and curved, cylindric, very finely velutinate under a lens, 1–4 mm long.

**Basidiospores** (8–)9.5–13(–14) × 6–8.5(–9.6) μm, L = 11.38 μm, W = 7.26 μm, Q = 1.54–1.61 (*n* = 90/3), ellipsoid to broadly ellipsoid, hyaline, thin-walled, smooth, with some guttules, faintly IKI+, CB−. **Basidia** 28–46 × 9–12 μm, cylindric or clavate, contain some guttules, 2-spored, sterigmata 5–7 μm long; basidioles in shape similar to basidia, but slightly smaller. **Gloeocystidia** present at edges of pores, in hymenium and pileipellis,

slightly thick-walled, smooth, contents dense and yellow-orange, those in hymenium clavate, 35–43 × 9–16 μm; those at pore edges more or less the same size as those in pileipellis, clavate, subglobose or ventricose, 18–45 × 10–25 μm. **Acanthocysts** at edges of pores, in hymenium and pileipellis, slightly thick-walled, contents dense and yellow-orange, clavate to oblong, few pyriform-elongated to irregular shaped, 15–45 × 6–11 μm. **Tramal** hyphae subparallel along tubes, strongly gelatinized, partly branched, hyaline, thin-walled, some slightly inflated, 3–5(–8) μm in diameter. Pileipellis comprising a palisade of acanthocysts and gloeocystidia. Hyphae in stipe parallel along stipe, slightly thick-walled, some inflated, 3–5(–10) μm in diameter. Clamp connections absent.

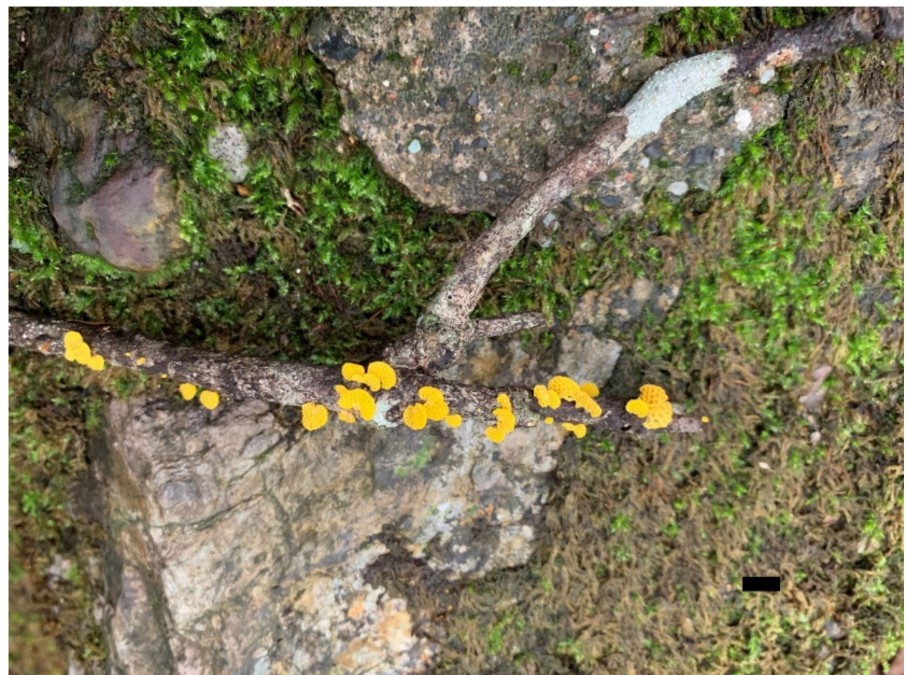

**Figure 5.** Fresh basidiocarps of *Favolaschia brevistipitata* (Holotype). Scale bar = 10 mm.

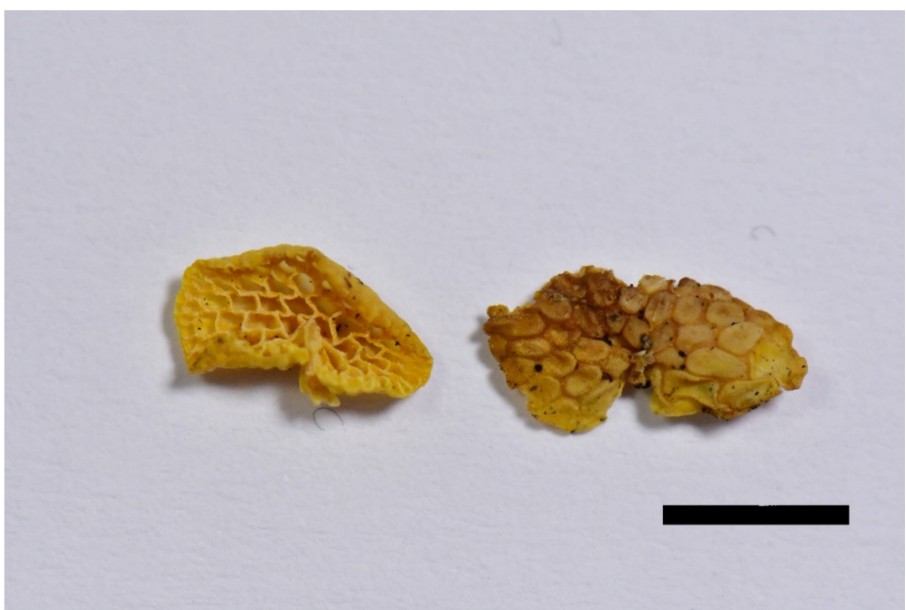

**Figure 6.** Dried basidiocarps of *Favolaschia brevistipitata* (Holotype). Scale bar = 5 mm.

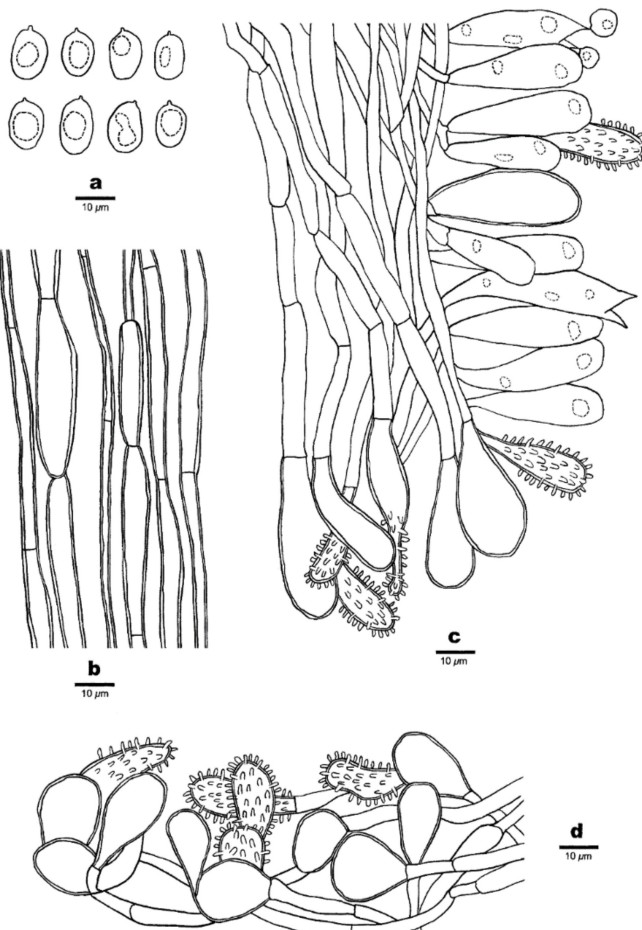

**Figure 7.** Microscopic structures of *Favolaschia brevistipitata* (Dai 19780, holotype). (**a**) Basidiospores. (**b**) Medullary hyphae of the stipe. (**c**) Hyphae at edge of pores showing gloeocystidia, acanthocystida, basidia and basidioles. (**d**) Hyphae of pileus cortex, composed of a hymeniform layer of acanthocysts and gloeocystidia.

Additional specimens (paratypes) examined. CHINA, Yunnan Province, Pingbian County, Daweishan National Forest Park, fallen angiosperm branch, 26 June 2019, Dai 19855 (BJFC031530), Dai 19856 (BJFC031531).

*Favolaschia claudopus* (Singer) Q.Y. Zhang & Y.C. Dai, stat. nov., Figures 8 and 9.

Basionym: *Favolaschia calocera* R. Heim var. *claudopus* Singer, Nova Hedwigia 50: 101, 1974—Holotype: New Zealand, Auckland, Waitakere Ranges, Waiatarua, on *Elaeagnus pungens*, July 1973, PDD 31006.

Epitype (designated here): Australia, Melbourne, Dandenong Ranges Botanical Garden, fallen trunk of *Eucalyptus*, 12 May 2018, Dai 18656 (BJFC027124, MEL).

MycoBank no.: MB 841449

**Diagnosis.** Differs from *F. brevibasidiata*, *F. brevistipitata*, *F. longistipitata*, and *F. minutissima* in larger pores. Differs from *F. calocera* in color.

**Basidiocarps** annual, gregarious, gelatinous. **Pileus** 4–14 × 3–10 mm, conchoid or reniform, lemon-chrome when fresh, becoming curry yellow when dry; pileal surface slightly undulate in a reticulate pattern matching the pores below, sometimes faintly pruinose when dry. **Hymenophore** concolorous with pileal surface, poroid, 30–170 pores per basidiocarp; pores 0.5–2.5 mm in diameter, pentagonal to hexagonal and sometimes irregularly elongated, larger near the base and smaller near the edge, the marginal pores often incomplete, pore edges pruinose when dry. **Stipe** obvious, laterally attached, concolorous with pileus, straight, cylindrical, or tapered to a slightly wider base, very finely velutinate under a lens, 3–10 mm long.

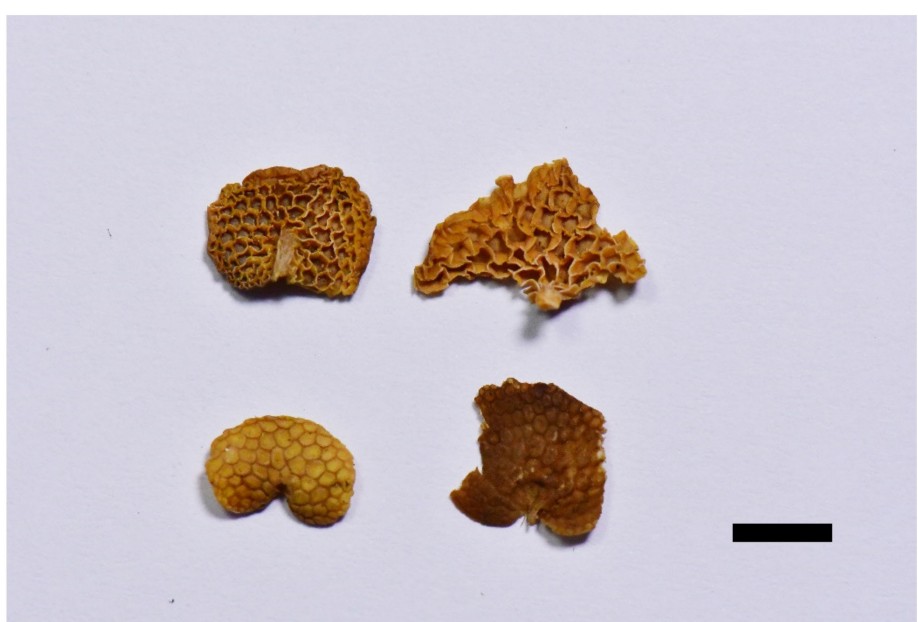

**Figure 8.** Dried basidiocarps of *Favolaschia claudopus* (Epitype). Scale bar = 5 mm.

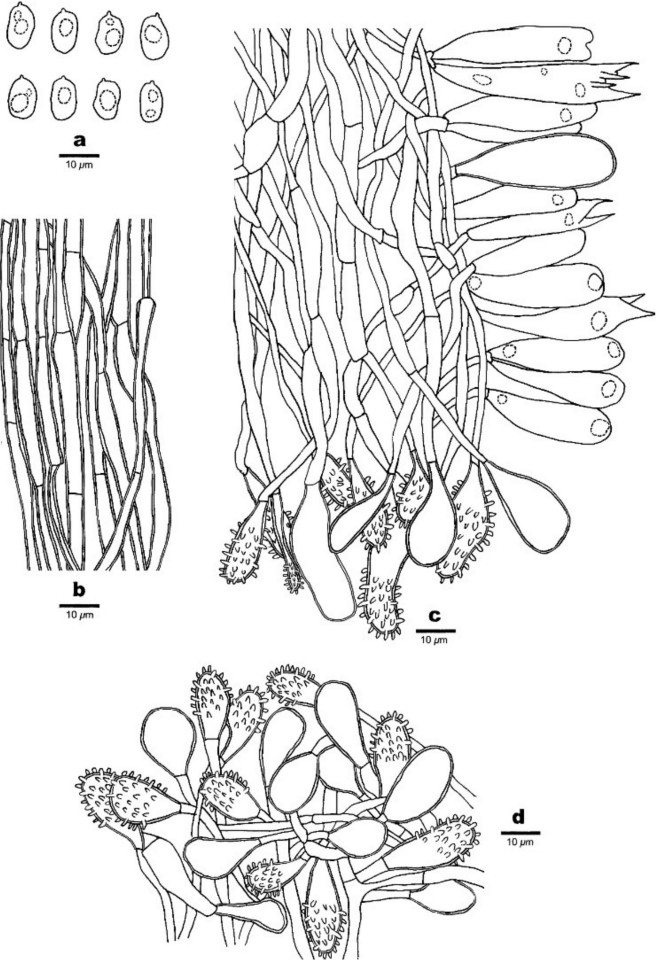

**Figure 9.** Microscopic structures of *Favolaschia claudopus* (Dai 18656, holotype). (**a**) Basidiospores. (**b**) Medullary hyphae of the stipe. (**c**) Hyphae at edge of pores showing gloeocystidia, acanthocystida, basidia and basidioles. (**d**) Hyphae of pleus cortex, composed of a hymeniform layer of acanthocysts and gloeocystidia.

**Basidiospores** (8–)9.8–15(–15.5) × 6–8.8(–9) μm, L = 12.11 μm, W = 7.12 μm, Q = 2.09–2.14 (*n* = 60/2), mostly ellipsoid or ovoid, few oblong, hyaline, thin-walled, smooth, with some guttules, faintly IKI+, CB−. **Basidia** 25–49 × 6–11 μm, cylindric or clavate, contain some guttules, 2(–4)–spored, sterigmata 6–9 μm long; basidioles in shape similar to basidia, but slightly smaller. **Gloeocystidia** present at the edges of pores, in hymenium and pileipellis, slightly thick-walled, smooth, contents dense and yellow-orange, those in hymenium ellipsoid or clavate, 35–50 × 10–14 μm; those at pore edges more or less the same size as those in pileipellis, clavate, subglobose to oblong, 17–52 × 8–26 μm. **Acanthocysts** at the edges of pores and in pileipellis, slightly thick-walled, contents dense and yellow-orange, clavate, subclavate with a constriction, sometimes elongated to irregular shaped, 15–47 × 6–12 μm. **Tramal** hyphae subparallel along tubes, strongly gelatinized, partly branched, hyaline, thin-walled, some slightly inflated, 2–6(–8) μm in diameter. Pileipellis comprising a palisade of acanthocysts and gloeocystidia. Hyphae in stipe parallel along stipe, slightly thick-walled, some slightly inflated, 2–4(–6) μm in diameter. Clamp connections absent.

Additional specimen examined. Australia, Melbourne, Dandenong Ranges Botanical Garden, fallen trunk of *Eucalyptus*, 12 May 2018, Dai 18663 (BJFC027131).

*Favolaschia longistipitata* Q.Y. Zhang & Y.C. Dai, sp. nov., Figures 10–12.

MycoBank no.: MB 841447.

**Diagnosis.** Differs from *F. brevibasidiata*, *F. calocera*, and *F. minutissima* in color. Differs from *F. brevistipitata* in longer stipes. Differs from *F. claudopus* in smaller pores.

**Type.** CHINA, Yunnan Province, Pingbian County, Daweishan National Forest Park, fallen angiosperm branch, 26 June 2019, Dai 19799 (holotype, BJFC031474).

**Etymology.** *Longistipitata* (Lat.): referring to the species having a long stipe.

**Basidiocarps** annual, gregarious, gelatinous. **Pileus** 2–12 × 1.5–8 mm, flabelliform to subcircular, lemon-chrome when fresh, becoming curry yellow when dry; pileal surface slightly undulated in a reticulate pattern matching the pores below, sometimes faintly pruinose when dry. **Hymenophore** concolorous with pileal surface, poroid, 40–240 pores per basidiocarp; pores 0.5–1.5 mm in diameter, pentagonal to hexagonal and sometimes irregularly elongated, larger near the base and smaller near the edge, the marginal pores often incomplete, pore edges pruinose when dry. **Stipe** obvious, laterally attached, concolorous with pileus, cylindrical or tapered to a slightly wider base, sometimes curved, very finely velutinate under a lens, 4–12 mm long.

**Basidiospores** (9–)9.8–13 × 6–8 μm, L = 11.31 μm, W = 6.72 μm, Q = 1.62–1.74 (*n* = 120/4), ellipsoid to oblong, hyaline, thin-walled, smooth, with some guttules, faintly IKI+, CB−. **Basidia** 32–47 × 9–13 μm, cylindric or clavate, few fusoid, contain some guttules, 2–spored, sterigmata 4–6 μm long; basidioles in shape similar to basidia, but slightly smaller. **Gloeocystidia** present at the edges of pores, in hymenium and pileipellis, slightly thick-walled, smooth, contents dense and yellow-orange, those in hymenium pyriform, 25–45 × 10–12 μm; those at pore edges more or less the same size as those in pileipellis, subglobose or ventricose, 20–40 × 6–21 μm. **Acanthocysts** at the edges of pores and in pileipellis, slightly thick-walled, contents dense and yellow-orange, elongated to irregular shaped, 16–42 × 5–13 μm. **Tramal** hyphae subparallel along tubes, strongly gelatinized, partly branched, hyaline, thin-walled, some slightly inflated, 2–4(–6) μm in diameter. Pileipellis comprising a palisade of acanthocysts and gloeocystidia. Hyphae in stipe parallel along stipe, slightly thick-walled, some slightly inflated, 2–4(–7) μm in diameter. Clamp connections absent.

Additional specimens (paratypes) examined. CHINA, Yunnan Province, Gengma County, Nangunhe Nature Reserve, on a fallen branch of *Castanopsis*, 11 July 2013, Dai 13226 (BJFC014716); Kunming, Xiaoshao Forest Farm, on a fallen angiosperm branch, 1 July 2019, Dai 20019 (BJFC031693); Lincang, Linxiang District, Xiaodaohe Forest Farm, on fallen branch of *Castanopsis*, 10 July 2013, Dai 13221(BJFC014711); Nanhua County, Dazhongshan Nature Reserve, on a fallen branch of *Quercus*, Cui 11128 (BJFC015243); Pingbian County, Daweishan National Forest Park, on a fallen angiosperm branch, 26 June

2019, Dai 19781 (BJFC031456), on a fallen trunk, 27 June 2019, Dai 19893 (BJFC031567); Wuding County, Shizishan Nature Reserve, on a fallen angiosperm trunk, 15 August 2019, Dai 20328 (BJFC031996), Dai 20355 (BJFC032023), on a fallen branch of *Alnus*, Dai 20341 (BJFC032009); Xinping County, Mopanshan Forest Park, on a fallen angiosperm branch, 15 May 2017, Dai 17597 (BJFC025129), Dai 17598 (BJFC025130), Dai 17601 (BJFC025133).

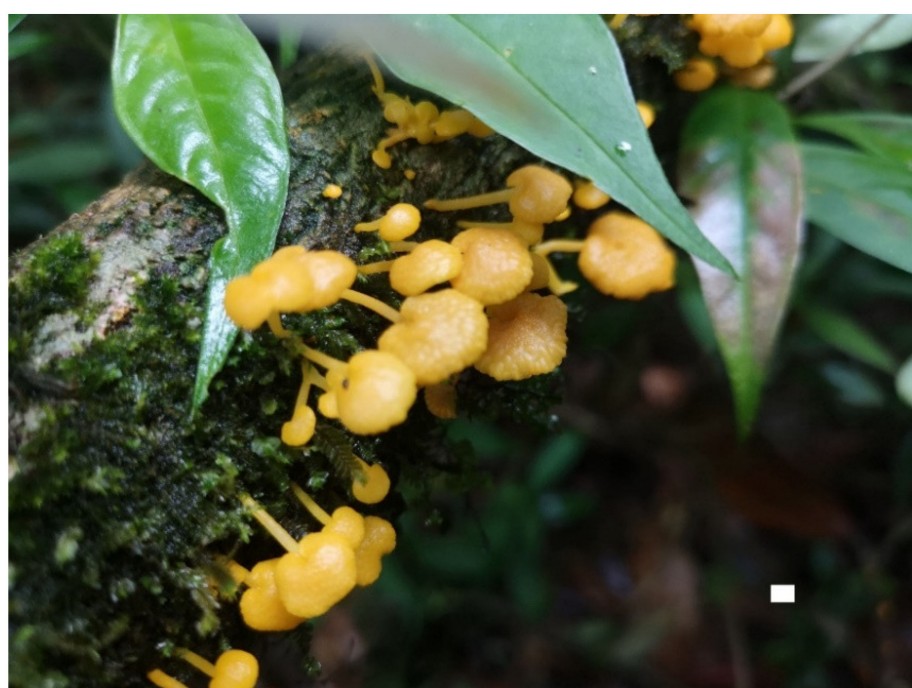

**Figure 10.** Fresh basidiocarps of *Favolaschia longistipitata* (Holotype). Scale bar = 10 mm.

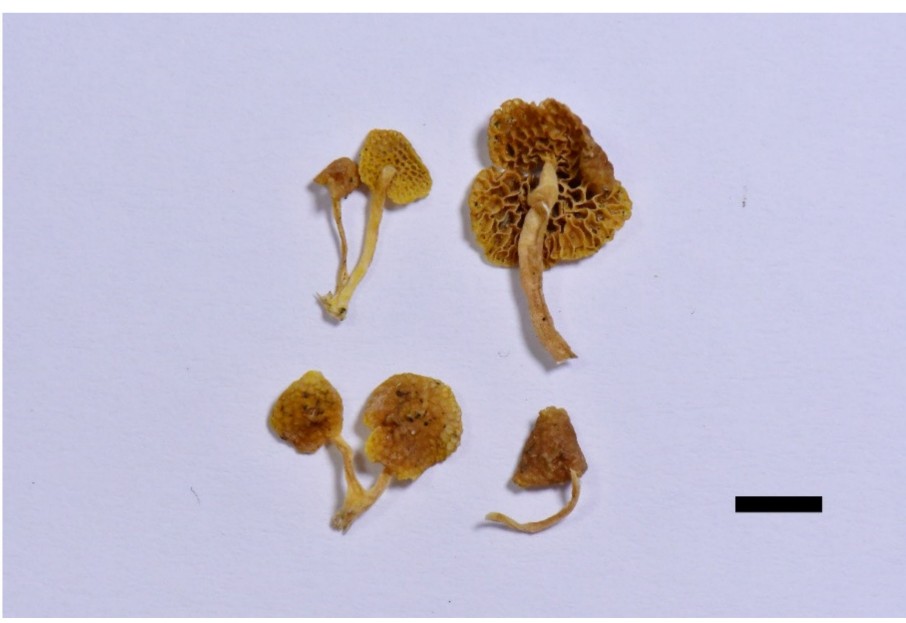

**Figure 11.** Dried basidiocarps of *Favolaschia longistipitata* (Holotype). Scale bar = 5 mm.

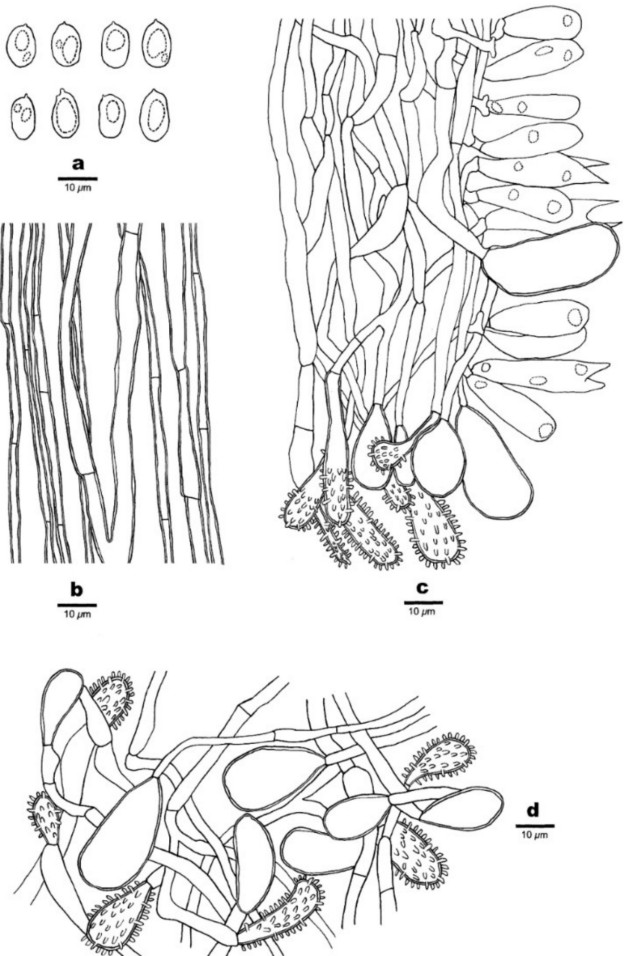

**Figure 12.** Microscopic structures of *Favolaschia longistipitata* (Dai 19799, holotype). (**a**) Basidiospores. (**b**) Medullary hyphae of the stipe. (**c**) Hyphae at the edge of pores showing gloeocystidia, acanthocystida, basidia, and basidioles. (**d**) Hyphae of the pileus cortex, composed of a hymeniform layer of acanthocysts and gloeocystidia.

*Favolaschia minutissima* Q.Y. Zhang & Y.C. Dai, sp. nov., Figures 13–15.
MycoBank no.: MB 841448.
**Diagnosis.** Differs from other *F. calocera* complex species in smaller basidiocarps and smaller pores.
**Type.** CHINA, Hainan Province, Ledong County, Jianfengling Naturel Reserve, fallen angiosperm branch, 3 July 2019, Dai 20086 (holotype, BJFC031760).
**Etymology.** *Minutissima* (Lat.): referring to the species having a very small basidiocarp.
**Basidiocarps** annual, gregarious, gelatinous. **Pileus** 2–4 × 1–3 mm, reniform to suborbicular, apricot-orange when fresh, becoming cream buff when dry; pileal surface slightly undulated in a reticulate pattern matching the pores below, sometimes faintly pruinose when dry. **Hymenophore** concolorous with pileal surface, poroid, 30–90 pores per basidiocarp; pores 0.2–0.5 mm in diameter, pentagonal to hexagonal and sometimes irregularly elongated, larger near the base and smaller near the edge, the marginal pores often incomplete, pore edges pruinose when dry. **Stipe** often scarcely developed, laterally attached, concolorous with pileus, short and curved, cylindric, very finely velutinate under a lens, 0.5–4 mm long.
**Basidiospores** (6–)7.5–11(–11.5) × (4.8–)5–7.2(–7.8) μm, L = 9.16 μm, W = 6.10 μm, Q = 1.46–1.56 (*n* = 90/3), broadly ellipsoid to ovoid, hyaline, thin-walled, smooth, with some guttules, faintly IKI+, CB−. **Basidia** 25–37 × 7–9.5 μm, clavate, few fusoid, contain some guttules, 2–spored, sterigmata 5–7 μm long; basidioles in shape similar to basidia, but

slightly smaller. **Gloeocystidia** present at the edges of pores, in hymenium and pileipellis, slightly thick-walled, smooth, contents dense and yellow-orange, those in hymenium clavate to vesiculose, 30–42 × 10–12.5 μm; those at the pore edges more or less the same size as those in pileipellis, pyriform, subglobose or ventricose, 20–48 × 10–26 μm. **Acanthocysts** at the edges of pores, in hymenium and pileipellis, slightly thick-walled, contents dense and yellow-orange, clavate to oblong, 9–45 × 7–12 μm. **Tramal** hyphae subparallel along tubes, strongly gelatinized, partly branched, hyaline, thin-walled, some slightly inflated, 2–4(–6) μm in diameter. Pileipellis comprising a palisade of acanthocysts and gloeocystidia. Hyphae in stipe parallel along stipe, slightly thick-walled, some slightly inflated, 2–4(–7) μm in diameter. Clamp connections absent.

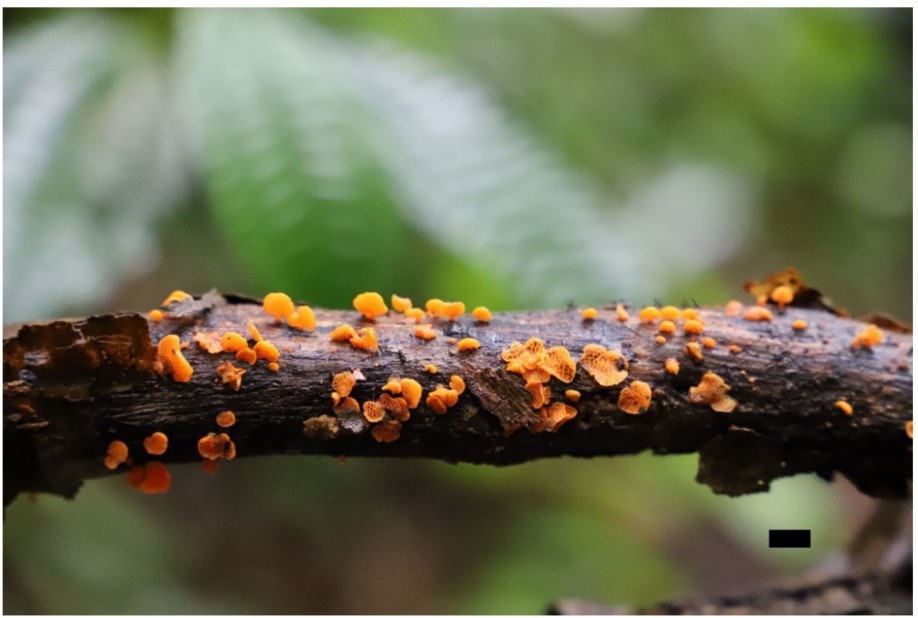

**Figure 13.** Fresh basidiocarps of *Favolaschia minutissima* (Holotype). Scale bar = 10 mm.

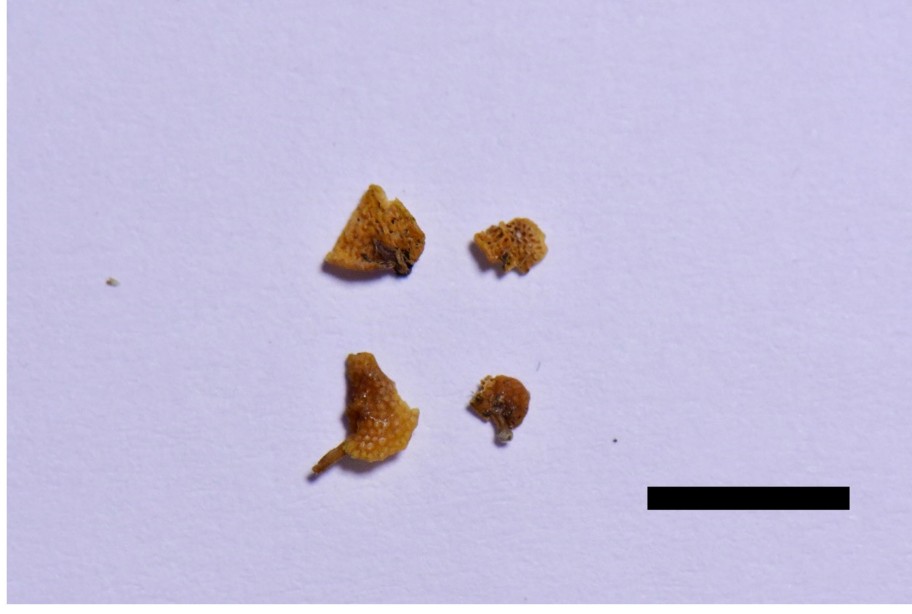

**Figure 14.** Dried basidiocarps of *Favolaschia minutissima* (Holotype). Scale bar = 5 mm.

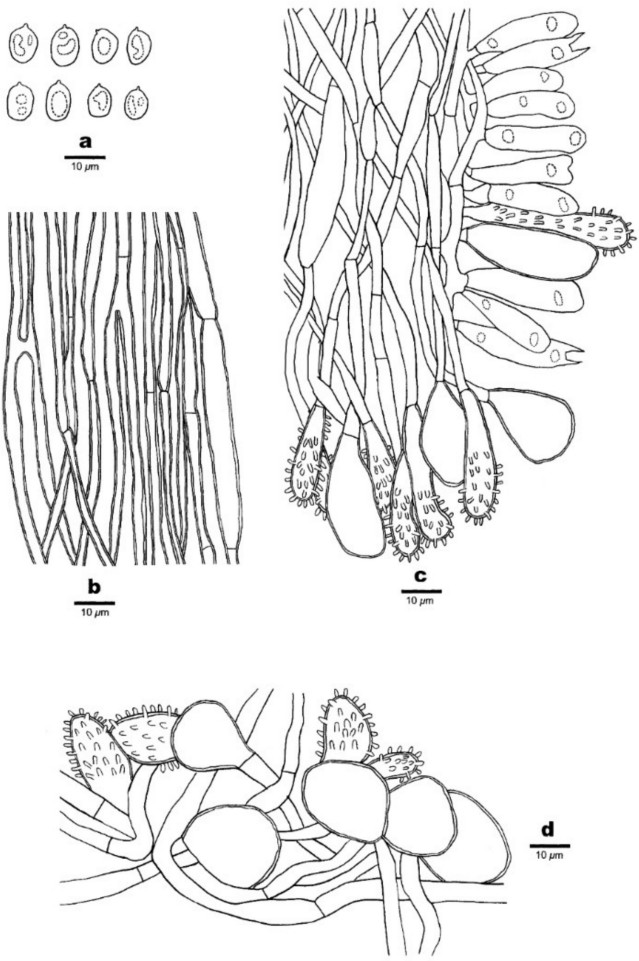

**Figure 15.** Microscopic structures of *Favolaschia minutissima* (Dai 20086, holotype). (**a**) Basidiospores. (**b**) Medullary hyphae of the stipe. (**c**) Hyphae at the edge of pores showing gloeocystidia, acanthocystida, basidia, and basidioles. (**d**) Hyphae of the pileus cortex, composed of a hymeniform layer of acanthocysts and gloeocystidia.

Additional specimens (paratypes) examined. CHINA, Hainan Province, Changjiang County, Bawangling Nature Reserve, a fallen angiosperm branch, 8 May 2009, Dai 10753 (BJFC004997); Ledong County, Jianfengling Nature Reserve, a fallen angiosperm branch, 3 July 2019, Dai 20085 (BJFC031759), Dai 20088 (BJFC031762).

### 4. Discussion

*Favolaschia calocera*, originally described in Madagascar, is a species complex. We recognize six species in the complex: *Favolaschia calocera sensu stricto*; three new species from China, *F. brevibasidiata*, *F. brevistipitata*, *F. longistipitata*; one new species from Asia, *F. minutissima*; and a variety raised to species rank *Favolaschia claudopus* (Singer) Q.Y. Zhang & Y.C. Dai from Oceania, Asia and Africa. Two samples of *Favolaschia calocera* from the type locality Madagascar form a separate clade; we refer to the description of Singer [8], which is adapted from Heim's original description and the sequence by Johnston et al. [13]. The members of *Favolaschia calocera* complex differ from other species in the genus by the bright orange or yellow basidiocarps with a distinct laterally stipe and numerous gloeocystidia and acanthocysts. The main morphological characteristics of species in *Favolaschia calocera* complex are listed in Table 2.

**Table 2.** A comparison of the characteristics of the species in the *Favolaschia calocera* complex.

| Species | F. brevibasidiata | F. brevistipitata | F. calocera | F. claudopus | F. longistipitata | F. minutissima |
|---|---|---|---|---|---|---|
| Basidiocarp size (mm) | 2–7 × 1.5–4 | 2–10 × 1–6 | 11 × 9 | 4–14 × 3–10 | 2–12 × 1.5–8 | 2–4 × 1–3 |
| Colour | apricot-orange when fresh, cream buff when dry | lemon-chrome when fresh, curry yellow when dry | orange | lemon-chrome when fresh, curry yellow when dry | lemon-chrome when fresh, curry yellow when dry | apricot-orange when fresh, cream buff when dry |
| Pores (diameter and number per basidiocarp) | 0.4–1 mm, 50–120 | 0.4–1.5 mm, 50–180 | up to 2 mm, 40–70 | 0.5–2.5 mm, 30–170 | 0.5–1.5 mm, 40–240 | 0.2–0.5 mm, 30–90 |
| Stipe (mm) | 2–8 | 1–4 | 9 | 3–10 | 4–12 | 0.5–4 |
| Basidia (µm) | 23–30 × 8–10 | 28–46 × 9–12 | 33–36 × 7–10 | 25–49 × 6–11 | 32–47 × 9–13 | 25–37 × 7–9.5 |
| Sterigmata number and length (µm) | 2, 1.5–4 | 2, 5–7 | 2, 10–12 | 2–4, 6–9 | 2, 4–6 | 2, 3–5 |
| Basidiospores (µm) | 10–12 × 6–7.8 | 9.5–13 × 6–8.5 | 12–12.5 × 8.2–9 | 9.8–15 × 6–8.8 | 9.8–13 × 6–8 | 7.5–11 × 5–7.2 |
| Locality | Hainan, China Yunnan, China | Yunnan, China | Madagascar | Australia, Italy, Kenya, New Zealand | Yunnan, China | Thailand Yunnan, China |

In the phylogenetic trees (Figures 1 and 2), our four new species and *Favolaschia claudopus* are nested in the *Favolaschia calocera* complex, and form five independent lineages. In addition, *Favolaschia brevibasidiata*, *F. calocera*, and *F. minutissima* are closely related. Morphologically, both *Favolaschia brevibasidiata*, *F. calocera*, and *F. minutissima* have apricot-orange pilei when fresh, but *F. calocera* differs from *F. brevibasidiata* by its larger basidiocarps (11 × 9 mm vs. 2–7 × 1.5–4 mm), larger pores (up to 2 mm vs. 0.4–1 mm), and longer basidia (33–36 × 7–10 µm vs. 23–30 × 8–10 µm). *Favolaschia minutissima* differs from *F. brevibasidiata* by its smaller basidiocarps (2–4 × 1–3 mm vs. 2–7 × 1.5–4 mm), smaller pores (0.2–0.5 mm vs. 0.4–1 mm), and smaller basidiospores (7.5–11 × 5–7.2 µm vs. 10–12 × 6–7.8 µm). Furthermore, *Favolaschia minutissima* is easily distinguishable from *F. calocera* by its smaller basidiocarps (2–4 × 1–3 mm vs. 11 × 9 mm), smaller pores (0.2–0.5 mm vs. up to 2 mm), smaller basidiospores (7.5–11 × 5–7.2 µm vs. 12–12.5 × 8.2–9 µm), and shorter stipes (0.5–4 mm vs. up to 9 mm). Moreover, *Favolaschia calocera* is distributed in Madagascar, but *F. brevibasidiata* and *F. minutissima* occur in Asia (southern China and Thailand).

*Favolaschia brevistipitata* occurs in Yunnan Province of China and forms a well-supported lineage (Figures 1 and 2). Morphologically, *Favolaschia brevistipitata*, *F. claudopus*, and *F. longistipitata* all have similar pilei (lemon-chrome when fresh, curry yellow when dry). However, *Favolaschia claudopus* differs from *F. brevistipitata* by its larger pores (0.5–2.5 mm vs. 0.4–1.5 mm) and longer stipes (3–10 mm vs. 1–4 mm). *Favolaschia longistipitata* can be easily separated from *F. brevistipitata* by its larger basidiocarps (2–12 × 1.5–8 mm vs. 2–10 × 1–6 mm) and longer stipes (4–12 mm vs. 1–4 mm). Microscopically, *Favolaschia brevistipitata* and *F. brevibasidiata* have similar sized basidiospores, but *F. brevibasidiata* differs from *F. brevistipitata* by its smaller basidiocarps (2–7 × 1.5–4 mm vs. 2–10 × 1–6 mm), longer stipes (2–8 mm vs. 1–4 mm), and smaller basidia (23–30 × 8–10 µm vs. 28–46 × 9–12 µm).

According to our study, two specimens of *Favolaschia claudopus* from Australia clustered together with specimens from Italy, Kenya, and New Zealand; however, our measurements on their basidiospores are slightly longer than those reported by Johnston et al. [11] (9.8–15 × 6–8.8 µm vs. 9–12.5 × 6.5–8.5 µm). In addition, *Favolaschia claudopus* can be easily separated from other species in *F. calocera* complex by its largest basidiocarps (4–14 × 3–10 mm) and largest pores (0.5–2.5 mm).

*Favolaschia longistipitata* is a common species in Yunnan Province of China. Phylogenetically, *Favolaschia longistipitata* is closely related to *Favolaschia claudopus* (Figure 1). Morphologically, *Favolaschia longistipitata* may be confused with *Favolaschia claudopus* in having lemon-chrome pileus when fresh, curry yellow when dry, and similar sized ba-

sidiocarps. However, *Favolaschia claudopus* differs from *F. longistipitata* in having larger pores (0.5–2.5 mm vs. 0.5–1.5 mm), longer gloeocystidia in pileipellis (17–52 × 8–26 μm vs. 20–40 × 6–21 μm), and longer sterigmata (6–9 μm vs. 4–6 μm). Both *Favolaschia longistipitata*, *F. brevibasidiata,* and *F. calocera* have similar sized basidiospores under the microscope. However, *Favolaschia brevibasidiata* is distinguished from *F. longistipitata* by its smaller basidiocarps (2–7 × 1.5–4 mm vs. 2–12 × 1.5–8 mm), shorter stipes (2–8 mm vs. 4–12 mm), and shorter basidia (23–30 × 8–10 μm vs. 32–47 × 9–13 μm). *Favolaschia calocera* differs from *F. longistipitata* by its a lesser pores per basidiocarp (40–70 vs. 40–240), shorter stipes (up to 9 mm vs. 4–12 mm), and longer sterigmata (10–12 μm vs. 4–6 μm).

Three newly sequenced samples *Favolaschia minutissima* with very small fruiting bodies from Hainan Province of China and one specimen from Thailand (DQ 026240, Johnston et al. 2006) formed a well-supported lineage (Figures 1 and 2). *Favolaschia minutissima* differs from other *F. calocera* complex species by its the smallest basidiocarps (2–4 × 1–3 mm), smallest pores (0.2–0.5 mm), and smallest basidiospores (L = 9.16 μm, W = 6.10 μm).

According to the molecular analysis performed by Johnston et al. [13] and Vizzini et al. [14], these collections of *Favolaschia calocera* from New Zealand, Kenya, and Italy clustered in one group represent very recent, probably human-vectored introductions; on the other hand, samples of Madagascar and Asia (southern China and Thailand) display a higher genetic variability and could represent the natural distribution of this fungus. Our study is consistent with the speculation that the Australian samples cluster with the specimens from New Zealand, Kenya, and Italy and are described as *Favolaschia claudopus*.

The natural distribution of *Favolaschia calocera* complex is a controversial question, which is being explored by taxonomists. Two hypotheses have emerged. One hypothesis suggested that *Favolaschia calocera* as well as other *Favolaschia* species, during the break-up of Gondwana (about 150 million years ago), drifted away from Madagascar, where they likely evolved, with the India–Seychelles landmass, and subsequently dispersed through Asia. A second hypothesis, contrary to the first, suggested that *Favolaschia calocera* was accidentally introduced into Madagascar from Asia, by an impressive one-way human migratory flow from Southeast Asia (Borneo) to Madagascar in the centuries between 200 and 500 Anno Domini [14,35–38]. As in our analysis, these Asian samples of *Favolaschia calocera* complex show higher variability than Madagascar. Our investigation revealed that *Favolaschia calocera* complex is common in tropical China and four new species have been discovered. However, to date, only *Favolaschia calocera* has been recorded in Madagascar in *F. calocera* complex, as mycologists have been active in Central America since the late1800s and it is unlikely they would have overlooked such a brightly colored and conspicuous fungus. Thus, our data tend to support the second hypothesis, but to confirm these hypotheses, a molecular clock analysis on all the *Favolaschia* species is needed.

Key to species of the *Favolaschia calocera* complex.
1. Basidiocarps apricot-orange when fresh................................................................................2
1. Basidiocarps lemon-chrome when fresh................................................................................4
2. Pileus usually <5 mm....................................................................................*F. minutissima*
2. Pileus usually >5 mm....................................................................................................3
3. Mature pores up to 2 mm in the largest dimension.................................................*F. calocera*
3. Mature pores 0.4–1 mm.............................................................................*F. brevibasidiata*
4. Stipe usually <5 mm long............................................................................*F. brevistipitata*
4. Stipe usually >5 mm long...........................................................................................5
5. Basidiospores up to 15 μm in length..............................................................*F. claudopus*
5. Basidiospores 9.8–13 × 6–8 μm...................................................................*F. longistipitata*

**Author Contributions:** Conceptualization, Y.-C.D. and Q.-Y.Z.; methodology, Q.-Y.Z.; perform the experiment, Q.-Y.Z.; formal analysis, Q.-Y.Z.; validation, Y.-C.D. and Q.-Y.Z.; investigation, Y.-C.D.; resources, Y.-C.D.; writing—original draft preparation, Q.-Y.Z.; writing—review and editing, Y.-C.D.; visualization, Q.-Y.Z.; supervision, Y.-C.D.; funding acquisition, Y.-C.D. All authors have read and agreed to the published version of the manuscript.

**Funding:** The research was supported by the Second Tibetan Plateau Scientific Expedition and Research Program (STEP), Grant No. 2019QZKK0503.

**Data Availability Statement:** Publicly available datasets were analyzed in this study. This data can be found here: https://www.ncbi.nlm.nih.gov/; https://www.mycobank.org; http://purl.org/phylo/treebase, submission ID 28714 (accessed on 30 August 2021).

**Conflicts of Interest:** The authors declare that there is no conflict of interest.

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
