# Peer review of "Taxonomy and Phylogeny of the Favolaschia calocera Complex (Mycenaceae) with Descriptions of Four New Species"

_forests, doi:10.3390/f12101397_

Round 1

Reviewer 1 Report

This manuscript presents some very nice phylogenetic and morphological analyses that delineate five new Asian species of Favolaschia in the F. calocera complex. The table of material used is excellent (but may need a repeating of the column headings on subsequent pages), and the synoptic key is very helpful.

 The submitted version, however, needs major revision.

First, the introduction and title need a broader phylogenetic/classification context. Nowhere is it mentioned that this genus is in the Mycenaceae. The family should also be included in parentheses in the title. Second, the authors repeat reports without questioning that these are saprotrophic. Many species of Favolaschia are host-specific, suggesting they may have a biotrophic stage. Many species of Mycena (Mycenaceae) are now known to be associated with plant roots (see Bugge Harder et al. 2021 https://doi.org/10.1101/2021.03.23.436563 and Thoen et al. 2020) In vitro evidence of root colonization suggests ecological versatility in the genus Mycena. New Phytol. The authors could indicate that Favolaschia produce basidiomes on dead plant material but that they may have a biotrophic phase as found in Mycena (Thoen et al. 2020, Bugge Harder et al. 2021) since many are known to be host-specific.

Second, one of the ‘new’ species names, Favolaschia claudopus Q.Y. Zhang & C. Dai, would lead to great confusion since the name has already been validly published at the rank of variety within the same species complex: Favolaschia calocera var. claudopus Singer. The Holotypes given are different – The Dai holotype is from Australia on Eucalyptus, whereas the Singer holotype is from New Zealand, PDD 31006, on branches of Elaeagnus pungens. Is this the same species? There is a different collection showing up in the sequences of F. claudopus from New Zealand, PDD75323. If PDD 31006 is the same, why have the authors not elevated Singer’s variety to species rank as Fav. claudopus (Singer) Q.Y. Zhang & C. Dai? If it was only to obtain a holotype with a full set of sequences, the best way would be to designate the Dai collection as an epitype rather than a holotype. If they are different species, this name should not be used. See Species Fungorum http://www.indexfungorum.org/names/NamesRecord.asp?RecordID=352810

Refer to the International Code of Nomenclature for algae, fungi, and plants 2018 Article 53.3, Ex. 15. https://www.iapt-taxon.org/nomen/pages/main/art_53.html#Art53.3

53.3. The names of two subdivisions of the same genus, or of two infraspecific taxa within the same species, even if they are at different ranks, are homonyms if they are not based on the same type and have the same final epithet, or are treated as homonyms if they have a confusingly similar final epithet. The later name is illegitimate.

Ex. 14. Andropogon sorghum subsp. halepensis (L.) Hack. (in Candolle & Candolle, Monogr. Phan. 6: 501. 1889) and A. sorghum var. halepensis (L.) Hack. (l.c.: 502. 1889) are legitimate because both have the same type (see also Rec. 26A.1).

Ex. 15. Anagallis arvensis subsp. caerulea Hartm. (Sv. Norsk Exc.-Fl.: 32. 1846), based on the later homonym A. caerulea Schreb. (Spic. Fl. Lips.: 5. 1771), is illegitimate because it is itself a later homonym of A. arvensis var. caerulea (L.) Gouan (Fl. Monsp.: 30. 1765), based on A. caerulea L. (Amoen. Acad. 4: 479. 1759).

Ex. 16. Scenedesmus armatus var. brevicaudatus (Hortob.) Pankow (in Arch. Protistenk. 132: 153. 1986), based on S. carinatus var. brevicaudatus Hortob. (in Acta Bot. Acad. Sci. Hung. 26: 318. 1981), is a later homonym of S. armatus f. brevicaudatus L. S. Péterfi (in Stud. Cercet. Biol. (Bucharest), Ser. Biol. Veg. 15: 25. 1963) even though the two names apply to taxa at different infraspecific ranks. However, S. armatus var. brevicaudatus (L. S. Péterfi) E. H. Hegew. (in Arch. Hydrobiol. Suppl. 60: 393. 1982) is not a later homonym because it is based on the same type as S. armatus f. brevicaudatus L. S. Péterfi.

Third, unless there are higher quality photographs of the basidiomes that were not included in the review copy, the resolution is poor. These are good for color, size and stipe attachment and length, but not much else. The drawings would need to be augmented with drawings of the basidiomes.

Fourth, the diagnoses are incorrect. The problem seems to be that the authors are giving distinguishing characters that differentiate among the species in the F. calocera complex, or that a combination of characters sets the species apart within the F. calocera complex, but the wording says that these features distinguish the new species from all other species of Favolaschia, which is not correct. F. brevistipitata says: “Diagnosis. Differs from other Favolaschia species by the lemon-chrome pileus when fresh” but F. claudopus, F. longistipitata and F. selloana also have a lemon-chrome pileus, and the first two are in the F. calocera complex! F. brevibasidiata is more or less the same color as F. variriotecta, and the short basidia, which is said to distinguish this species, is about the same length as in F. gaillardia, F. cinnabarina, F. selloana & F. varariotecta (15-29 µm). They are, however, shorter than in the other species of the F. calocera complex. The discussion of characters that distinguish the species from one another in this species complex are good, and could serve as models (with reduced wording) for the diagnoses (e.g., Differs from F. x in __ and __. Differs from F. y in __).

Lines 51-52 This does not belong in the introduction. It can be in the abstract, results and discussion, but not in the last paragraph of introduction: “it proved to a species 51 complex, based on morphological and phylogenetic studies.”

There are also a moderate number of problems with wording or word endings.

Lines 53, “The aim of this study is to investigate the diversity and phylogeny within the Favolaschia calocera species complex,

Lines 54-55, change to: Five new species are illustrated and described, and key characteristics for distinguishing species are provided.

Line 59 says specimens are deposited in Beijing, but it should say specimens collected by the authors or new collections from China, Australia and Thailand, or new collections from ….

Line 62, change ‘were’ to ‘are’

Lines 94 and 95, insert ‘the’ in front of ‘outgroup’

Line 142 change “distinctly form six lineages” to “form six distinct lineages”

Line 143 change “into” to “in” and italicize Favolaschia calocera

Lines 634-635 “Favolaschia calocera, originally described in Madagascar, is a species complex, six species are recognized in the complex” Break into two sentences as: “Favolaschia calocera, originally described in Madagascar, is a species complex. We recognize six species in the complex, …..”

Author Response

Point 1: 
the introduction and title need a broader phylogenetic/classification context. Nowhere is it mentioned that this genus is in the Mycenaceae. The family should also be included in parentheses in the title. Second, the authors repeat reports without questioning that these are saprotrophic. Many species of Favolaschia are host-specific, suggesting they may have a biotrophic stage. Many species of Mycena (Mycenaceae) are now known to be associated with plant roots (see Bugge Harder et al. 2021 https://doi.org/10.1101/2021.03.23.436563 and Thoen et al. 2020) In vitro evidence of root colonization suggests ecological versatility in the genus Mycena. New Phytol. The authors could indicate that Favolaschia produce basidiomes on dead plant material but that they may have a biotrophic phase as found in Mycena (Thoen et al. 2020, Bugge Harder et al. 2021) since many are known to be host-specific.

Response 1: We have modified the introduction and title, which are reflected in the manuscripts.

Point 2: one of the ‘new’ species names, Favolaschia claudopus Q.Y. Zhang & C. Dai, would lead to great confusion since the name has already been validly published at the rank of variety within the same species complex: Favolaschia calocera var. claudopus Singer. The Holotypes given are different – The Dai holotype is from Australia on Eucalyptus, whereas the Singer holotype is from New Zealand, PDD 31006, on branches of Elaeagnus pungens. Is this the same species? There is a different collection showing up in the sequences of F. claudopus from New Zealand, PDD75323. If PDD 31006 is the same, why have the authors not elevated Singer’s variety to species rank as Fav. claudopus (Singer) Q.Y. Zhang & C. Dai? If it was only to obtain a holotype with a full set of sequences, the best way would be to designate the Dai collection as an epitype rather than a holotype. If they are different species, this name should not be used. See Species Fungorum http://www.indexfungorum.org/names/NamesRecord.asp?RecordID=352810

Response 2: Thanks!

We treated the taxon as following:

Favolaschia claudopus (Singer) Q.Y. Zhang & C. Dai, comb. et stat. nov.

Basionym: Favolaschia calocera var. claudopus Singer, Beih. Nova Hedwigia 50: 101 (1974)

Holotype: New Zealand. Auckland, Waitakere Ranges, Waiatarua, on Elaeagnus pungens, R. E. Beever, Jul 1973, PDD 31006.

Epitype (designated here): Australia, Melbourne, Dandenong Ranges Botanical Gar-den, fallen trunk of Eucalyptus, 12 May 2018, Dai 18656 (BJFC027124, MEL).

Point 3: Third, unless there are higher quality photographs of the basidiomes that were not included in the review copy, the resolution is poor. These are good for color, size and stipe attachment and length, but not much else. The drawings would need to be augmented with drawings of the basidiomes.

Response 3: We re-treated the photos, and their qualities are good enough.

Point 4: Fourth, the diagnoses are incorrect. The problem seems to be that the authors are giving distinguishing characters that differentiate among the species in the F. calocera complex, or that a combination of characters sets the species apart within the F. calocera complex, but the wording says that these features distinguish the new species from all other species of Favolaschia, which is not correct. F. brevistipitata says: “Diagnosis. Differs from other Favolaschia species by the lemon-chrome pileus when fresh” but F. claudopus, F. longistipitata and F. selloana also have a lemon-chrome pileus, and the first two are in the F. calocera complex! F. brevibasidiata is more or less the same color as F. variriotecta, and the short basidia, which is said to distinguish this species, is about the same length as in F. gaillardia, F. cinnabarina, F. selloana & F. varariotecta (15-29 µm). They are, however, shorter than in the other species of the F. calocera complex. The discussion of characters that distinguish the species from one another in this species complex are good, and could serve as models (with reduced wording) for the diagnoses (e.g., Differs from F. x in __ and __. Differs from F. y in _).

Response 4: The diagnoses are revised, which are reflected in the manuscripts.

Point 5: Lines 51-52 This does not belong in the introduction. It can be in the abstract, results and discussion, but not in the last paragraph of introduction: “it proved to a species 51 complex, based on morphological and phylogenetic studies.”

Response 5: Thank you for your advice, because this is already described in abstract and discussion, we deleted the sentence.

Point 6: Lines 53, “The aim of this study is to investigate the diversity and phylogeny within the Favolaschia calocera species complex,

Response 6: It is done.

Point 7: Lines 54-55, change to: Five new species are illustrated and described, and key characteristics for distinguishing species are provided.

Response 7: It is done.

Point 8: Line 59 says specimens are deposited in Beijing, but it should say specimens collected by the authors or new collections from China, Australia and Thailand, or new collections from ….

Response 8: The sentence was revised.

Point 9: Line 62, change ‘were’ to ‘are’

Response 9: It is done.

Point 10: Lines 94 and 95, insert ‘the’ in front of ‘outgroup’

Response 10: It is done.

Point 11: Line 142 change “distinctly form six lineages” to “form six distinct lineages”

Response 11: It is done.

Point 12: Line 143 change “into” to “in” and italicize Favolaschia calocera

Response 12: It is done.

Point 13: Lines 364-365 “Favolaschia calocera, originally described in Madagascar, is a species complex, six species are recognized in the complex” Break into two sentences as: “Favolaschia calocera, originally described in Madagascar, is a species complex. We recognize six species in the complex, …..”

Response 13: The sentence is revised.

Reviewer 2 Report

Most of my comments concern corrections to English.  See the attached document with suggested changes.

Below are some more specific comments.

  1. Abstract.  Some verbs could be past tense.  Maybe the editor can decide what is best.
  2. Pileus and stipe dimensions. You give only one dimension, like for brevibasidiata - 7 x 4 mm, but more than one basdiome was collected (??) If so, I would suggest giving a range, e.g., 5-7 x 4-5 mm. Same for the stipe, there should be a range of dimensions.
  3. Plates. I would recommend cropping photos B, E, G so they can be enlarged.
  4. Table.  The table can be shortened by deleting some of the characters that do not differentiate the species, like tramal hyphae diameter, stipe hyphae.
  5. Key.  If possible include more than one distinguishing feature in the keys, maybe spore size, basidiocarp color, etc.
  6. Discussion. The discussion is difficult and tedious to read because there are so many comparisons.  I would highly recommend that a discussion be presented after each species description.  That way the reader can focus only on that species and what it is related to.  It would probably not be that much work to move the content of he discussion to the individual species.  The final paragraphs of the discussion could remain in the discussion.
  7. This manuscript will be a valuable contribution to our knowledge of Favolaschia. 

Author Response

Point 1: Lines 8. described from Madagascar . . .

Response 1: It is done.

Point 2: Lines 9. change to distributed worldwide; what is meant by “America”, if Panama is the country, then use Central America

Response 2: The sentence is revised.

Point 3 Lines 10. . . . were analyzed. . . I think past tense is better in the abstract but the editor can decide if necessary.

Response 3: It is done.

Point 4: Lines 11. were based on . . . .

Response 4: It is done.

Point 5: Lines 14. (nu-SSU), and the translation.

Response 5: It is done.

Point 6: Lines 16. was discovered from  . . .; were found

Response 6: It is done.

Point 7:. Lines 17. delete namely

Response 7: It is done.

Point 8: Lines 19. characterisitics of species in the Favolaschia. . . .

Response 8: It is done.

Point 9: Lines 27. represent only around 50 species only

Response 9: It is done.

Point 10: Lines 32. occur in different .

Response 10: It is done.

Point 11: Lines 33. forests; bush does not make sense, in English it is not an environment type, do you mean shrub-lands?

Response 11: The sentence is revised.

Point 12: Lines 34. and occurs on over 50 . . . .

Response 12: It is done.

Point 13: Lines 41. it has been spread into Europe.

Response 13: It is done.

Point 14: Lines 42. gradually diverged

Response 14: It is done.

Point 15: Lines 43. Differences

Response 15: It is done.

Point 16: Lines 49. Displays

Response 16: It is done.

Point 17: Lines 50. from Central America

Response 17: It is done.

Point 18: Lines 51. Thailand were analyzed and it proved . . .

Response 18: It is done.

Point 19: Lines 54. new species are found described.

Response 19: It is done.

Point 20: Lines 59. Did you actually study specimens from other herbaria? 

Response 20: We performed both molecular and morphological studies on the specimen from Institute of Microbiology, Beijing Forestry University (BJFC), and phylogenies based on samples from America, Australia, China, Kenya, Italy, New Zealand and Thailand are analyzed.

Point 21: Lines 141. based on ITS, (insert comma after ITS); shows that the five .. . ; italicize Favolaschia calocera and line 143, F. calocera.

Response 21: It is done.

Point 22: Lines 165. only one pileus dimension is given – 7 x 4; do you not have other basidiomes to include in the dimensions, so it would read something like 5-7 x 4-5, etc.  And see other descriptions.

Response 22: We have modified the description of basidiomes, which is reflected in Taxonomy, Discussion and Table 2.

Point 23: Lines 172. is there a range for stipe length? And see other descriptions.

Response 23: We have modified the description of stipe length, which is reflected in Taxonomy, Discussion and Table 2.

Point 24: Lines 190. Are A, C, D, F. H dried basidiomes.  If so indicate in caption.  I would suggest cropping photos B, E, and G., so that the photos so that they can be enlarged in the manuscript.

Response 24: We have modified the photos, which are reflected in the manuscripts.

Point 25: Lines 211-212. finely velutinous – is there anything microscopically that is associated with the velutinous condition? And see other descriptions.

Response 25: Velutum can be easily observed in the body-type microscopy.

Point 26: Lines 238. big larger basidiocarps

Response 26: It is done.

Point 27: Lines 241. etymology for claudopus (“limp foot”)

Response 27: etymology is not needed for the taxon Favolaschia claudopus (Singer) Q.Y. Zhang & C. Dai

Point 28: Table 2.: the table could be shortened by deleting some of the characters that do not reveal anything taxonomically, e.g., Tramal hyphae diameter, Stipe hyphae, possibly even Gloeocystidia and Acanthocyst measurements. 

Response 28: It is done.

Point 29: Lines 365. one new species from Australia, F. claudopus; four new species from China, F. . . . . . . (move the from Australia and from China to before the species names)

Response 29: The sentence is revised.

Point 30: Lines 388. China, and forms a well- supported. . ..

Response 30: The sentence is revised

Point 31: Lines 392. The mean spore lengths are 11.3 μm for brevistipitata and 12.11 μm  for claudopus – that is hardly a difference and may not be statistically significant.

Response 31: The sentence is deleted.

Point 32: Lines 430-431.  Sentence is awkward, re-word: question, it is being explored by taxonomists with two hypotheses emerging.  One hypothesis . . . .

Response 32: The sentence is revised.

Point 33: Lines 441-442, re-word: to date, only Favoloaschia calocera, from the calocera complex, has been recorded from Madagascar.  Mycologists have been active in America;  need to define America (N. America, S. America, Central America, etc.)

Response 33: The sentence is revised.

Point 34: Lines 444-445. re-word: hypotheses, but to confirm these hypotheses, a molecular clock analysis is needed for all species of Favolaschia.

Response 34: It is done.

Point 35: Key, lines 446-456. 

Couplet 3.  The table states that the pores of calocera are up to 2 mm, the key says less than 2 mm.
Couplet 4. The is a color difference as well: claudopus – lemon chrome, calocera – orange

Try to use more than one in character in the key if possible.

Response 35: The key is revised.

Point 36: Discussion. The discussion is difficult and tedious to read because there are so many comparisons. I would highly recommend that a discussion be presented after each species description. That way the reader can focus only on that species and what it is related to. It would probably not be that much work to move the content of the discussion to the individual species. The final paragraphs of the discussion could remain in the discussion.

Response 36: We prefer to keep the discussion in the present model, otherwise we have to repeat the similar comparison in the five new taxa.

Round 2

Reviewer 1 Report

The edits to the originally submitted manuscript are very positive improvements.

There are a few minor points for final revision of the manuscript.

In the abstract, lines 18-19, consider rewording ‘is erected as an independent species.’ as ‘raised to species rank.’ While it is correct as written, the alternative wording may be better.

Line 38, delete ‘inhabit’ – the words ‘occurs on’ replace it.

All of the species diagnoses have been greatly improved, and the addition of the key to species in the complex is very helpful. The second member of each numbered couplet in the key should be distinguished with a mark, and the species or number references on the right should be right-margin-justified.

The additional photos are also a great help in distinguishing the species.

Line 243, there is an error in the statement ‘Favolaschia claudopus (Singer) Q.Y. Zhang & Y.C. Dai, comb. et stat nova, Figures 8, 9’ – REMOVE ‘comb.’ from ‘comb. et stat. nova’ so it only has ‘stat. nova’. For this to be a ‘comb. nov.’ the basionym would have to have been in a different genus, but it isn’t. The basionym is Favolaschia calocera R. Heim var. claudopus Singer’.

Line 250 – the MycoBank number (required for publication) is missing, but the MycoBank record needs to be corrected by removing 'comb. nova'.

Line 391 in discussion also needs to be corrected from ‘and a new combination’ to ‘and a variety raised to species rank’.
